# Preventive Pathways for Healthy Ageing: A Systematic Literature Review

**DOI:** 10.3390/geriatrics10010031

**Published:** 2025-02-18

**Authors:** Alice Masini, Niccolò Cherasco, Andrea Conti, Irlanda Pighini, Francesco Barone-Adesi, Massimiliano Panella

**Affiliations:** 1Department of Translational Medicine, Università del Piemonte Orientale, 28100 Novara, Italy; alice.masini@uniupo.it (A.M.); 20051926@studenti.uniupo.it (N.C.); 10025977@studenti.uniupo.it (I.P.); francesco.baroneadesi@uniupo.it (F.B.-A.); massimiliano.panella@med.uniupo.it (M.P.); 2Doctoral Program in Food, Health, and Longevity, Università del Piemonte Orientale, 28100 Novara, Italy

**Keywords:** healthy ageing, preventive pathways, lifestyle interventions, health promotion, active ageing, systematic literature review, elderly

## Abstract

**Background**: The world’s population is not only growing but also ageing, and healthcare systems should adapt to the needs of an ageing population. Until now, there has been no clear definition of a preventive pathway with the aim of improving lifestyles and promoting healthy and active ageing. The present systematic review aims to provide evidence to support the development of effective ways of delivering preventive pathways for healthy ageing. **Methods**: Several databases were searched, i.e., MEDLINE, COCHRANE, CINAHL, and PsycINFO, by using specific inclusion criteria, such as elderly population (i.e., subjects aged 65 years and older), preventive interventions for healthy ageing, studies with or without control groups, and effectiveness and methodological structure of the prevention pathway. The risk of bias was assessed by using the Joanna Briggs Institute and mixed methods appraisal tools. **Results**: A total of 9998 studies were identified after the removal of duplicates, and after screening title, abstracts, and full text, 14 studies were finally included. All the prevention pathways described are based on physical activity (PA) programmes, dietary interventions, and cognitive and mental health. The professional figures involved in the pathways were experts in prevention and health promotion, like family and community nurses, kinesiologists, and experts in stress management. The majority of the preventive pathways were implemented in primary care and community settings. **Conclusions**: Our systematic review provides evidence for developing an effective preventive healthy ageing pathway through tailored PA, diet, and cognitive health interventions. This co-designed approach should involve a multidisciplinary expert team and be implemented in primary care and community settings to improve psycho-physical health and longevity.

## 1. Introduction

Health promotion and healthy ageing have become imperatives today, reflecting a major shift in our society towards an emphasis on healthy lifestyles and prevention. In recent years, and in response to this need, the World Health Organization (WHO) announced that the concept of health promotion is the most effective way to promote human health [1]. In fact, the world’s population is not only growing; it is also ageing. This phenomenon is known as the “third demographic transition”, and it is estimated that by 2050, there will be an increase in the number of people aged 65 and over, i.e., the elderly [2]. The European Commission estimates that by 2060, the proportion of European citizens aged over 65 will increase from 18% to 28% and the proportion aged over 80 will increase from 5% to 12% (European Commission, 2015). Ageing is a process that although it is intrinsic to human nature, is influenced by a complex interaction among biological, cultural, community, and environmental aspects. The WHO defines active ageing as “the process of optimising opportunities for health, participation, and security in order to enhance quality of life as people age” for “helping people stay in charge of their own lives for as long as possible as they age and, where possible, to contribute to the economy and society” [3].

The demographic transition towards an aging population presents multifaceted consequences that not only affect individual health trajectories but also pose substantial socioeconomic challenges and put strain on healthcare delivery systems. To address these challenges, more than 190 countries endorsed the Global Strategy and Action Plan on Ageing and Health (2016–2030). This plan aims to promote healthy ageing, defined as “the process of developing and maintaining the functional ability (i.e., people’s capabilities of being and doing what they have reason to value) that enables well-being in older age” [4]. Therefore, the goal of healthy ageing should be primarily focused on improving physical, psychological, and mental health in the population aged 60 and over [5].

In consideration of the ageing process, we need to take into account that frailty-related conditions such as chronic diseases are rising. This trend creates new health needs for the population and requires a shift from an acute care model to a coordinated and comprehensive continuum of care. Healthcare systems need to adapt to this change and reorganise themselves, as current models of care are inadequate to meet the health needs of a rapidly ageing population [6].

The WHO urges member states to base their quality improvement policies on the entire continuum of care, taking into account the criteria of effectiveness, safety, equity, efficiency, integration of care, and timeliness [7]. To this end, a continuum of care should serve to reduce medical costs while providing optimal patient-centred care [8]. A care pathway is a complex intervention for the mutual decision making and organisation of care processes for a well-defined group of patients during a well-defined period [9]. The majority of care pathways are tailored for a specific disease. To date, no clear definition of a preventive pathway aimed to improve active and healthy ageing exists. According to the WHO, living arrangements including social support, social wealth, and background factors (e.g., age, gender, marital status, employment status, educational background, income, and size of the family) are considered influential in active and healthy ageing [10].

In order to improve both longevity and quality of life in older adults, the lifestyle approach is one of the most promising strategies for enhancing longevity and quality of life in older adults [5]. Lifestyle factors are key targets for behaviour change interventions focused on promoting healthy ageing with PA, nutrition, and cognitive function [11].

Therefore, our study is aimed at supporting the development of preventive pathways for healthy ageing through a systematic review of the most recent scientific evidence.

## 2. Materials and Methods

### 2.1. Search Strategy

This systematic review was conducted in accordance with the PRISMA guidelines [12]. The study protocol is available in the International Prospective Register of Systematic Reviews—PROSPERO (ID: CRD42023431045). The search strings were developed considering the following PICOS (Patients, Interventions, Comparators, Outcomes, and Study design): (P) older adults, namely, subjects aged 65 years or over; (I) preventive pathways or preventive programs for healthy ageing; (C) both studies with and without control groups were included; (O) efficacy and methodological structure of the existing preventive pathway or preventive programs; (S) primary research based on original data. Detailed PICOS criteria are available in Table 1.

### 2.2. Selection Criteria

The literature search was conducted in June 2023 on MEDLINE (PubMed), Cochrane Central Register of Controlled Trials, Cumulative Index to Nursing and Allied Health Literature (CINAHL), Psychological Information Database (PsycINFO) (EBSCO), and Scopus. We searched electronic databases, with a 10-year publication date limit, because we were interested in recent approaches. Search strategies (strings adapted when necessary to fit the specific search requirements of each database) used the following keywords combined with Boolean Expressions (i.e., queries using logical operators “AND”, “OR” and “NOT”, between the main terms): *((Pathway* OR Path OR Approach) AND (“Preventive Health Services”[Majr] OR Care Preventive Health OR Health Care Preventive OR Preventive Care OR Preventive Health OR Health Preventive OR Preventive Health Programs OR Health Program Preventive OR Health Programs Preventive OR Preventive Health Program OR Program Preventive Health OR Programs Preventive Health OR Preventive Programs OR Preventive Program OR Program Preventive OR Programs Preventive) AND (“Aged”[Mesh] OR elderly) AND (“Methods”[Mesh] OR methodology OR barrier* OR facilitator* OR Design OR Model* OR Modeling OR Model* composition OR Current organization) NOT (laboratory))*. Filters applied: Full text; published in the last 10 years; Humans; English; Aged: 65+ years, 80, and over (80+ years). The inclusion criteria were as follows: (1) language: articles written in English; (2) study design: both experimental and observational studies with original primary data; (3) population of interest: older adults and the elderly with mean age >65 years with or with no chronic conditions; (4) intervention: preventive pathways or preventive programs for healthy ageing; (5) outcome measurement: efficacy, and methodological structure and characteristics of the existing preventive pathway or preventive programs; (6) comparison: not relevant, both studies with or without control groups were included. The exclusion criteria were as follows: (1) articles not pertinent to the research topic; (2) populations of a different age (adult mean age <65 years); (3) studies with interventions also including other topics; (4) study protocols or other papers without original data. Moreover, we conducted a grey literature search of other papers, using hand searches of key conference proceedings, journals, professional organisations’ websites, and guideline clearing houses. Finally, with a snowball technique, we examined references cited in the primary papers to identify additional eligible papers.

### 2.3. Data Extraction

After the removal of duplicates performed by adopting Zotero (Roy Rosenzweig Center for History and New Media; www.zotero.org/; (accessed on 24 October 2024)), title and abstract screening was performed with Microsoft Excel (Microsoft Corp, Redmond, PA, USA). This was conducted by a total of three researchers. Specifically, two researchers (N.C. and I.P.) screened the records, while the third one (A.C.) was consulted in case of disagreement. The research team contacted the study authors in the case where more information was deemed necessary. The same approach was adopted for full-text screening and data extraction. The retrieved information was inserted in a Google Sheet database, which included the following fields: name of the first author, publication year, country, study design, population details (including age and gender), intervention description (including setting, duration, and measured outcomes), and results. Data extraction followed the methods provided by the Cochrane Reviewers’ Handbook [13]. The data extraction phase was performed manually, and the content of each included paper was thematically analysed.

### 2.4. Risk of Bias Assessment

To assess the risk of bias for the potentially included studies, two reviewers (A.C. and I.P.) independently and blindly applied the Joanna Briggs Institute (JBI) critical appraisal instruments [14,15,16] and the mixed methods appraisal tool (MMAT) [17]. The tiebreaker (A.M.) was consulted in case of conflict. The risk of bias evaluation was made based on the primary outcome of interest: methodological structure and characteristics of the existing preventive pathway or preventive program, according to the PRISMA guidelines [12]. The JBI checklist for randomized controlled trial (RCT) studies [14] comprises thirteen items designed to evaluate different methodological aspects, including the randomisation and the allocation concealment process, the blindness of participants and researchers, and the statistical analysis used. Items four and five (“Were participants blind to treatment assignment?”; “Were those delivering the treatment blind to treatment assignment”) were not considered, due to the nature of the included studies which investigated lifestyle intervention. The JBI checklist for quasi-experimental (QES) studies [15] comprises nine items designed to evaluate different methodological aspects of this kind of studies, including the methods, the outcomes, the characteristics of the participants, the description of the treatment, control, follow-up, and the statistical analysis used. The JBI checklist for qualitative studies (QSs) [16] comprises ten items designed to evaluate different methodological aspects of this kind of studies, including five items for assessing research methodology congruity (i.e., stated philosophical perspective, research question or objectives, methods used to collect data, and representation and analysis of data). Each checklist presents different possible answers: “yes”, “no”, “unclear”, or “not applicable”. The MMAT scale [17] is designed for the appraisal stage of mixed methods studies (MMSs) comprising two parts, where the first part has questions related to qualitative design and the second part for quantitative design. The criteria to generate the overall score were decided and approved by all the authors. In detail, the overall score was assigned as follows: (i) “High quality” if all the criteria were met. (ii) “Medium quality” if at least one domain was unclear. (iii) “Low quality” if at least one domain was not met.

## 3. Results

### 3.1. General Characteristics

A total of 9998 unique records were retrieved after duplicate removal. A total of 261 articles were passed in the abstract screening phase. Finally, 14 studies fully meeting the eligibility criteria were included in this systematic review (Figure 1). The characteristics of the studies are shown in Table 2.

The geographic origins of the study were as follows: United Kingdom (n = 1) [19], United States (n = 1) [27], The Netherlands (n = 3) [21,22,29], Canada (n = 1) [31], Germany (n = 2) [18,26], Japan (n = 1) [28], China (n = 3) [24,25,30], Iran (n = 1) [20], and Australia (n = 1) [23]. The studies were published between 2015 and 2023. The study designs were RCT = 7 [18,19,20,25,28,29,30], quasi-experimental = 3 [21,24,27], qualitative = 2 [23,26], and mixed methods = 2 [22,31]. The age of the population ranged from 60 to 85 years old, while the sample size varied from 34 to 986 participants. The study duration ranged from 12 weeks to 12 months.

### 3.2. Risk of Bias Results

Table 3 summarises the quality of the included studies assessed with different scales. The complete quality appraisal is available in the Appendix A.

With regards to the randomised control trials, both Davodi et al. [20] and Beyer et al. [18] were scored as being of “low quality” due to the absence of a clear randomisation process, blindness of assessors, and follow-up data. A medium-quality result was obtained by three studies [19,25,29]. Finally, two studies [28,30] obtained a high level of quality due to the correct explanation for all domains. As concerns the quasi-experimental studies, only one study [21] obtained a high level due to the correct answers given in each evaluation field, while Schwingel et al. [27] was assessed as being of low quality due to the absence of a control group and follow-up, and one study [24] was rated as being of medium quality for the unclear follow-up description [23,26]. The qualitative study performed by Green et al. [23] obtained a medium-quality rating due to the lack of clarity regarding the researcher’s cultural or theoretical positioning, while the other qualitative study [26] was scored as being of high quality. Finally, the mixed methods studies conducted by Franse et al. [22] and by Yusupov et al. [31] obtained medium quality due to the unclear aspect of the quantitative components of the study, such as the completed outcome data and the representative sample size.

### 3.3. Preventive Pathway Characteristics

The 14 included studies reported detailed information related to the preventive pathway interventions implemented during their project. The complete data extraction is shown in Appendix B. The preventive pathways examined in the included studies were based on various specific lifestyle interventions aimed at healthy aging.

#### 3.3.1. Physical Activity Component

The majority of the preventive pathways included a fundamental component of PA [18,19,20,21,22,23,24,25,27,28,29]. The main difference among the studies was self-managed PA [19,24,27,28,30] compared with other studies in which the activities were totally supervised by professionals or researchers [18,20,21,22,25,29]. The PA program was delivered in both a supervised manner and a non-supervised manner, like in the “Moving well” component of the “Being your Best” preventive pathway conducted by Green et al., in which multicomponent PA was delivered through group fitness activities in the community and educational booklets for home-based intervention [23]. The supervised PA programs were tailored for older adults and seniors and designed to suit the needs and the physical abilities of this age group. They are crucial to maintaining or improving mobility, muscle strength, and balance, which are key aspects of reducing the risk of falls. Particularly, refs. [22,27] included a specific PA program to reduce falls in the elderly. Self-managed PA varied from studies in which a booklet with exercises was given to the participants, who managed on their own all the activities [19,24], to other studies where a counselling process was implemented [27,30] in order to monitor adherence to the intervention, and in other studies, participants were invited to join extra PA group activities in the territory. In the study conducted by [19], each participant could decide the amount of time to devote to specific activities in order to be more realistic in developing an approach that could potentially be disseminated at a population level [19]. Moreover, the “Agewell” preventive pathway performed by [19] compared two experimental groups: one group with mentoring during the activities and the other with just goal setting. The most important exercise components of PA intervention, both in supervised and non-supervised activities, included in the pathway were balance exercises, strength exercises, aerobic activities (i.e., walking, swimming, or cycling), and flexibility. These exercises are often combined into a comprehensive program that also takes into consideration individuals’ pre-existing medical conditions, personal preferences, and health goals. All the 12 studies [18,19,20,21,22,23,24,25,27,28,29,30] which included PA programs in the preventive pathway showed important effects of increasing PA levels, preventing injuries and illnesses, and promoting a sense of overall well-being and self-esteem, encouraging older adults to lead an active and fulfilling lifestyle. The results of the included studies showed an improvement in physical health outcomes assessed with different objectives, such as actigraphs, fitness tests (senior fitness test, physical performance battery, and gait speed), steps counts, Falls Efficacy Scale International (FES-I) and self-reported measurements like the PA Scale for the Elderly PASE, sedentary behaviours (hours per day), regular exercise (number of sessions per week), body composition (i.e., BMI, body fat percentage), blood sample (i.e., cholesterol levels), cardiovascular risk, and interviews.

#### 3.3.2. Dietary Component

A total of nine studies [19,20,23,24,25,27,28,29,30] included a dietary intervention in the preventive pathway. The characteristics of this dietary component varied from nutritional general advice to booklets with diet recommendations or small group lessons to ensure that participants received a balanced diet that contributes to the prevention of chronic diseases and the maintenance of optimal health [19,20,23,24,25,27,28,30]. Only in the “SLIMMER” preventive pathway, a personalised program (i.e., a specific diet routine) was implemented for all the participants [29]. The effectiveness of the dietary components of the preventive pathway described were assessed by using specific questionnaires for adherence to the Mediterranean diet, like Medas, Kidmed, and food frequency questionnaires, which underlined a significant effect of adopting a more healthy diet and increasing adherence to the Mediterranean diet.

#### 3.3.3. Cognitive Training and Mental Health

Aspects of cognitive training for healthy ageing were included in nine studies [18,19,20,23,24,25,27,28,31]. The type of activities implemented varied from memory training, social gatherings, and group therapy to support mental health, better deal with the ageing process, and prevent isolation, which can negatively impact overall health. Moreover, two studies [24,28] implemented fundamental training related to digital knowledge, the internet, and social networks to obtain a positive effect on digital health literacy. Cognitive and executive improvements were obtained in the majority of the studies [18,19,20,23,24,25,27,28,31]. Several instruments were implemented in order to assess cognitive and executive functions (i.e., the Montreal Cognitive Assessment (MoCA), the Florida Cognitive Activities Scale (FCAS), and the California Verbal Learning Test (CVLT)). Moreover, depressive symptoms and psychological well-being were monitored (i.e., the German Version of the Center of Epidemiological Studies Depression Scale (CED-D 10), the Geriatric Depression Scale (GDS), and the General Self-Efficacy Scale (GSE)). Yusopov et al. is the only included study who based the preventive pathway for healthy ageing entirely on an online cognitive training programme called “memory and ageing”. The researchers involved the elderly in a previous qualitative phase in order to have their feedback to develop the app [31]. Among the cognitive and mental health outcomes, three studies were focused specifically on stress [20,24,27], proposing coping strategies and counselling activities in order to improve stress management during ageing.

#### 3.3.4. Other Characteristics

Among the included studies, other interventions were included in the preventive pathway. Refs. [24,30] incorporated the fundamental component of financial security in their preventive pathway in order to improve the economic skills of the elderly. The two studies conducted by Franse et al. [21,22] included a fundamental component related to polypharmacy and loneliness in the “UHCE” preventive pathway. The “UHCE” program was designed according to a bottom-up approach starting from a qualitative design. Refs. [24,25] included important chronic disease prevention and management support and training in the preventive pathway.

Finally, ref. [26] created the "Reaching the elderly" preventive pathway totally based on home visits to address the specific needs of older adults in their home environment. This pathway was designed according to feedback obtained from semi-structured interviews and focus groups. Lastly, ref. [20] was the only one that included spiritual aspects in the preventive pathway as fundamental to the ageing process of the Iranian population.

### 3.4. Professional Figures and Settings of Preventive Pathways

Several professional figures managed the preventive pathway components. First of all medical doctors, general practitioners, physicians, and experts in geriatrics coordinated many preventive pathways, in particular the enrolment and the evaluation phase to start the program [20,21,22,23,24,25,29,30]. Nurses were fundamental to the assessment phase and counselling activities in different programs [21,22,23,25,29]. The prevention pathways involved new professionals—kinesiologists and exercise trainers/experts—who collaborated with physiotherapists to manage exercise and PA components [18,20,21,22,24,25,28,29].

The cognitive and mental health components were managed by a psychologist or expert in mental health only in five studies [18,20,21,22,24]. Finally, the component relating to a healthy diet was managed by nutrition experts in only three studies [25,26,29]. Other fundamental professionals were involved in the care pathways, like social workers, social scientists, pharmacists, volunteers, e-learning designers, and experts in family relations and in social protection [25,26,29]. The only two studies who never mentioned in detail the professional figures involved in the program and just mentioned “research teams” were Clare et al. [19] and Yusopov et al. [31]. The study conducted by Schwingel A. et al. [27] involved a specific figure in the pathway and called them “Promotoras”. The researchers recruited and trained older women in the community for about 18 h throughout nine training modules that lasted about 2 h each. In several studies, religion was used as a means of facilitating community participation in prevention. For example, in the study by Davodi et al. [20], the inclusion of spiritual aspects was considered fundamental to the ageing process of the Iranian population, while in the study by Schwingel [27], the church and priests were the first point of contact with the Latino community and the starting point for implementing the programme. As concerns the setting of the preventive pathways, in the study conducted by [26], “Reaching the elderly” was the only programme based on home visits. The majority of the preventive pathways were performed in primary healthcare settings [21,22,25,29], community dwellings [18,23,27,31], or community centres [19,24,28,30]. Finally, ref. [20] conducted their “Active ageing” preventive pathway in a health and treatment centre.

## 4. Discussion

### 4.1. Main Results: Preventive Pathway Characteristics

The present is the first systematic review summarising 14 recent studies related to preventive pathways for healthy ageing. Overall, the present systematic review provides evidence to develop an effective preventive pathway for healthy ageing for older adults and the elderly. Figure 2 summarises the main characteristics of a preventive pathway for healthy aging.

Starting from the characteristics of the preventive pathways, the majority of the included studies focused the preventive program on lifestyle interventions, particularly PA, nutrition, and cognitive and mental health. These characteristics are in line with the aim of prevention and health promotion for healthy ageing proposed by the WHO [32] and the recent literature, which proves how lifestyle intervention represents a best buy in public health [5]. A growing body of literature suggests that the improvement in the health of older adults will not be achieved through a healthy lifestyle [33,34,35]. PA, nutrition, cognitive enhancement, stress management, and social engagement represent the main pillars of lifestyle medicine [36,37]. Moreover, the most effective preventive pathways for healthy ageing included in our systematic review pay attention to the social and psychological needs of the elderly [18,19,20,21,22,23,25,26,27,28,29,30]. Considering that older adults and the elderly have complex needs, the healthcare system should be adapted to meet their requirements, especially since the demand will increase as the population ages. These results are in line with the recent guidelines about healthy ageing priorities proposed by the WHO [38]. In fact, most studies have not only created a tailored program aimed at improving lifestyles but have also tried to take into consideration the motivation and compliance of the subjects involved. The personalisation of the interventions to meet these needs is essential and unique to older adults, highlighting how well-designed interventions can actively improve ageing, increase involvement in healthy activities, and above all, mitigate risk factors for chronic diseases [27]. In the majority of the included studies, the implemented design takes into account theoretical models of changed behaviour [39] or different behavioural models, like the socioecological model [40], on which to base programs in order to improve adherence and compliance. Precisely for this reason, several preventive programs were based on PA and cognitive intervention led by professionals but also tasks and activities that can be handled independently by the individual to facilitate adherence to and the management of the pathway without excessive limitations [19]. It is in this scenario that making the pathways accessible online represents an innovative aspect to be considered, like in the study performed by Yusopov [31], which takes into consideration the use of technology to spread these good healthy ageing practices. In addition, the results of our systematic review underline how the participation of older adults and the elderly in the creation of a preventive pathway is fundamental to its effectiveness and sustainability over time. In fact, among the included studies, the majority have envisaged a co-design phase of the intervention or a qualitative assessment [19,21,23,25,26,27,29,30,31], thus adopting a qualitative study design in which the interested stakeholders were involved to best co-design the pathways based on the needs that emerged. Incorporating the perspectives of the elderly and care providers is essential to establishing principles that make community health centres more age-friendly and better equipped to meet the needs of patients and the communities they serve [41].

### 4.2. Professional Figures and Settings

The adoption of these preventive pathways, which have demonstrated significant effectiveness in terms of improving psycho-physical well-being and quality of life, also presents numerous challenges, including cultural and linguistic barriers, which can undermine effectiveness in the elderly community. This trend was observed in studies conducted in low–middle-income countries [20,27]. For example, in the study by [27], Latino participants expressed mistrust towards research, society, and the healthcare system. This mistrust can negatively influence participation in health promotion programs. Promoting healthy lifestyles among these populations requires a culturally sensitive approach. This is the reason why the intercession of the so-called “promotoras” (community leaders) was fundamental and proved effective, as they spoke the local language and understood the culture of the participants. The “promotoras” receive specific training to transmit health information in an accessible and understandable manner. Moreover, in low–middle-income countries, religious experts were involved in the preventive pathway as one of the main contact points in order to improve adherence to the program. Furthermore, the present systematic review emphasises the growing necessity of involving new professional figures in preventive pathways. This includes not only medical and nursing staff, especially the emerging role of community and family nurses [42] in dealing with chronic conditions, but also experts in prevention and health promotion who collaborated together. Notably, professionals such as kinesiologists are becoming increasingly significant as physical exercise specialists, particularly in programs conducted outside traditional healthcare settings. For older adults, it is important to begin any new exercise program under the guidance of qualified professionals, such as physiotherapists and kinesiologists or graduates in sports science specialising in training older adults, to ensure that exercises are performed safely and with the right intensity even when self-managed. As regards the management of the healthy diet component, a different situation emerges. In most preventive programs, this component is managed by various professional figures who are not specifically trained on the topic. Only the studies conducted by Van Dongen al., Patzel et al., and Lee et al. enrolled an expert in healthy diet and nutrition in the multidisciplinary team [25,26,29]. Generally, the results of this systematic review underline the importance of continuing organisational support, clinical champions who communicate regularly with healthcare professionals, dedicated staffing, and ongoing data collection to easily and timely implement appropriate interventions. As regards the implementation setting, it is well recognised that addressing the growing burden of chronic diseases necessitates opportunities for health promotion and disease prevention within the community, as well as effective disease management within healthcare services. Several chronic diseases and related disabilities that could affect a person’s life span, along with their economic and human costs, can be prevented. However, prevention means that action needs to be taken before the start of the disease or before the disease takes hold, and that means taking a life course approach to active and healthy ageing [41]. Primary healthcare centre settings and community dwellings represents the most frequent environments in which the majority of preventive healthcare and screening for early disease detection and management take place. These primary healthcare centres, to which people can self-refer, also provide the bulk of ongoing management and care. It is estimated that 80% of front-line healthcare is provided at the community level, with primary healthcare serving as the foundation of the healthcare system [41]. Integrated care aims to transition from inpatient to ambulatory and outpatient care, emphasising home-based interventions, community engagement, and a well-coordinated referral system, as described in the majority of the included studies. This integration should happen at the healthcare organisation and community levels, as well as within policies, economic mechanisms, and shared governance structures. Furthermore, an integrated care system involves diverse professional figures operating in a variety of settings [43].

### 4.3. Limitations

This systematic review presents some limitations. First of all, we only searched for articles related to preventive pathways for the elderly aged over 65 years old from the last 10 years, since we were interested in recent approaches. The recent literature suggests that the healthy ageing process and new interventions should start even before then, during childhood and adolescence. Even if this systematic review followed the current methodology [12], it is worth mentioning that the search string terms might not be completely exhaustive and could have led us to miss some studies. Moreover, the included studies presented several heterogeneity. First of all, some studies have relatively small samples [19,23], which may limit the generality of the results. A larger sample could provide more robust and representative data. The variability in the methods and study designs produces some difficulties in the direct comparison of the results. Finally, the insufficient long-time follow-up limits the understanding of the lasting effects of interventions. It is essential to include prolonged follow-ups to evaluate the sustainability of the benefits of the interventions.

## 5. Conclusions

The present systematic review provides evidence to develop an effective preventive pathway for healthy ageing characterised by tailored co-design PA, healthy diet, and cognitive health components considering the levels of self-perceived motivation to change lifestyle. The future so-called “preventive pathway for healthy ageing” should present a multidisciplinary task force with several professionals and experts in health promotion and prevention. Preventive pathways should be implemented in primary care and community settings involving local community leaders in the development phase in order to be effective in improving psycho-physical health and increase longevity.

## Figures and Tables

**Figure 1 geriatrics-10-00031-f001:**
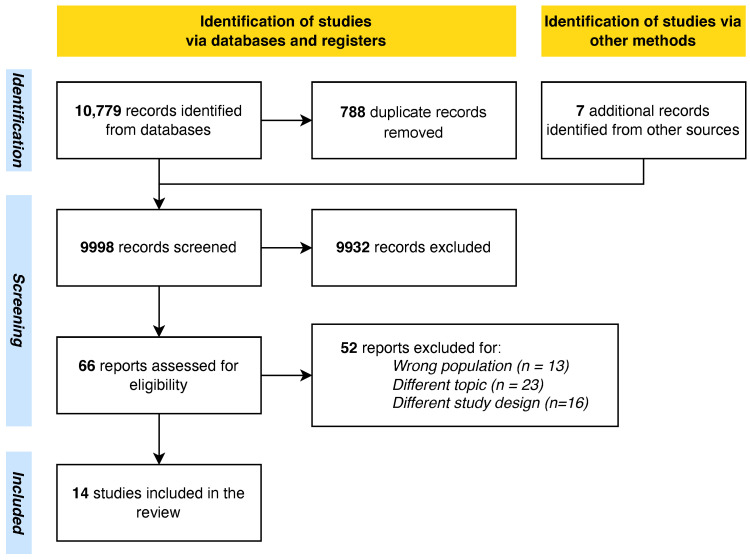
PRISMA flowchart.

**Figure 2 geriatrics-10-00031-f002:**
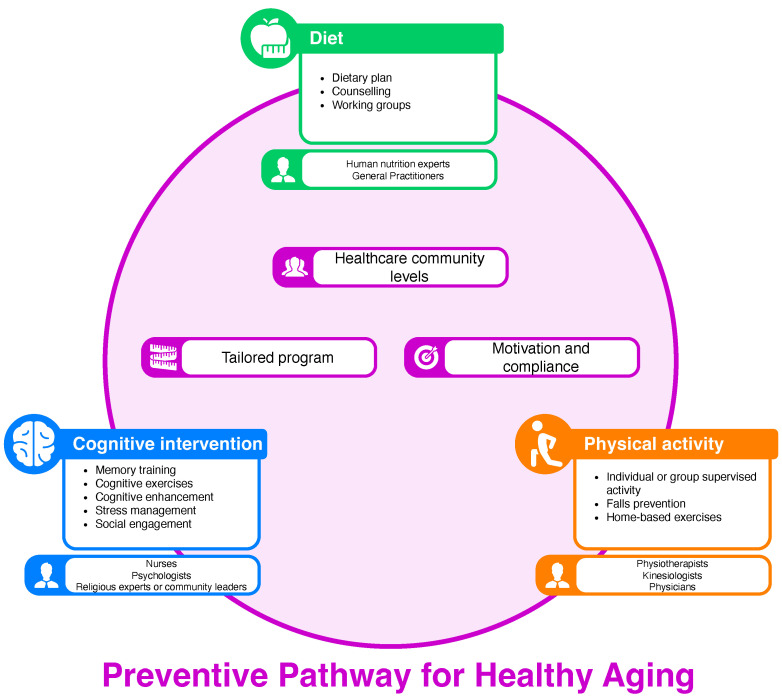
Preventive pathway characteristics.

**Table 1 geriatrics-10-00031-t001:** PICOS eligibility criteria.

Parameter	Inclusion Criteria	Exclusion Criteria
Population	Older adult/elderly * (i.e., subject aged 65 years and older) with or without chronic conditions	Adult mean age lower than <65 years
Intervention	Preventive pathways or preventive programs for healthy aging	Critical care pathway
Comparator	N/A	N/A
Outcome	Methodological structure and characteristics of the existing preventive pathway or preventive programs	
Study design	Experimental (i.e., RCTs, quasi-experimental studies, and pilot studies), observational study (i.e., cohort studies, longitudinal studies, and qualitative studies), or document papers describing the methodology of quasi-experimental or observational studies. Studies with full-text written in English.	Systematic reviews, meta analyses, or other papers without original data (i.e., reviews, letters to editors, editorials, book chapters, and conference abstracts).

* The terms “older adult” and “elderly” are used as synonyms in this review.

**Table 2 geriatrics-10-00031-t002:** Characteristics of the included studies.

Study and Year	Country	Study Design	Subjects (n)	Age (Mean)	Gender (Female, %)	Setting
Beyer et al., 2019 [18]	Germany	RCT	Total: 84; CG: 38; EG: 46.	Total: 76.8 ± 5.29.	65.2	Community dwelling
Clare L. et al., 2015 [19]	United Kingdom	RCT	Total: 75; IG: 27; GSG: 24; GSM: 24.	Total: 68.21; IG: 70.22 ± 7.77; GSG: 67.50 ± 7.66; GSM: 68.21 ± 7.92.	Total: 86.7; IG: 85.2; GSG: 95; GSM: 79.2.	Community centre
Davodi et al., 2023 [20]	Iran	RCT	Total: 60; CG: 30; EG: 30.	CG: 78.16 ± 7.07; EG: 66.5 ± 5.1	CG: 56.7; EG: 46.6.	Health and treatment centre
Franse et al., 2018 [21]	The Netherlands	Quasi-experimental	Total: 1844; CG 858; EG: 986.	Total: 79.5 ± 5.6; CG: 79.7 ± 5.5EG: 79.3 ± 5.7.	Total: 60.8; CG: 61.4; EG: 60.3.	Primary care and community settings in five European cities
Franse al. 2019 [22]	Germany	Mixed methods	Total: 986; FCP: 278; PCP: 130; LCP: 223; MCP: 94.	Total: >75 years.	EG1: 70.1; EG2: 50.8; EG3: 71.3; EG4: 62.8.	Primary care and community settings
Green et al. 2021 al. [23]	Australia	Qualitative	Total: 40; consumers: 23; professionals: 17.	Consumers: 69.3 ± 15.	Consumers: 65.	Community dwelling
Hsu et al., 2018 [24]	Taiwan	Quasi-experimental	Total: 123; CG: 43; EG person to person: 61; EG person to digital: 54.	CG: 78.16 ± 7.07; EG person to person: 77.25 ± 8.27; EG person to digital: 76.57 ± 7.01.	CG: 71.9; EG: 74.1.	Community care centres
Lee et al., 2021 [25]	Taiwan	RCT	Total: 398; EG: 199; CG: 199.	EG: 73.2 ± 6.1; CG: 73.1 ± 6.5.	EG: 62.0; CG: 58.0.	Primary healthcare setting
Patzelt C. et al., 2016 [26]	Germany	Qualitative	Total: 42.	Total: 65 and older.	Total: 52.4.	Community settings
Schwingel A. et al., 2017 [27]	USA	Quasi-experimental	Total: 34.	Total: 64 ± 8.	N/A	Community dwelling
Uemura et al. 2020 [28]	Japan	RCT	Total: 60; EG: 30; CG: 30.	EG: 74.0 ± 4.9; CG: 73.9 ± 4.4.	EG: 66.6; CG: 66.6.	Rural community
van Dongen et al., 2016 [29]	The Netherlands	RCT	Total: 316; EG: 155; CG: 161.	EG: 60.7 ± 6.4; CG: 61.0 ± 6.5.	EG: 47.7; CG: 50.3.	Dutch primary healthcare
Wong et.al, 2022 [30]	China	RCT	Total: 92; EG: 46; CG: 46.	Total: 75.9 ± 7.8; EG: 73.7 ± 6.9; CG: 78.2 ± 8.1.	Total: 78.0; EG: 84.8; CG: 84.8.	Community centres from a non-governmental organisation
Yusupov et al., 2022 [31]	Canada	Mixed methods	Total: 40.	N/A	N/A	Community dwelling

CG: control group; EG: experimental group; RCT: randomised controlled trial; IG: information group; GSG: goal-setting group; GSM: goal-setting with mentoring group; FCP: falls care pathway; PCP: polypharmacy care pathway; LCP: loneliness care pathway; MCP: frailty/medical care pathway.

**Table 3 geriatrics-10-00031-t003:** Quality assessment of the included studies.

Authors	Study Design	Tool for Assessment	Overall Quality
Beyer et al. [18]	RCT	JBI for RCT	Low
Clare et al. [19]	RCT	JBI for RCT	Medium
Davodi et al. [20]	RCT	JBI for RCT	Low
Franse et al. [21]	QES	JBI for QES	High
Franse et al. [22]	MMS	MMAT	Medium
Green et al. [23]	QS	JBI for QS	Medium
Hsu et al. [24]	QES	JBI for QES	Medium
Lee et al. [25]	RCT	JBI for RCT	Medium
Patzelt et al. [26]	QS	JBI for QS	High
Schwingel et al. [27]	QES	JBI for QES	Low
Uemura et al. [28]	RCT	JBI for RCT	High
Van Dongen et al. [29]	RCT	JBI for RCT	Medium
Wong et al. [30]	RCT	JBI for RCT	High
Yusupov et al. [31]	MMS	MMAT	Medium

## Data Availability

Data are contained within the article and the Appendix A.

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
