# Peer review of "Preventive Pathways for Healthy Ageing: A Systematic Literature Review"

_geriatrics, 2025, doi:10.3390/geriatrics10010031_

Round 1

Reviewer 1 Report

Comments and Suggestions for Authors

Comments on the Quality of English Language

English is good. Some little parts could be however improved.

Author Response

Dear Reviewer, thank you so much for the insightful comments and suggestions provided. We improved the manuscript according to all the Reviewers' comments. Please see attached our response to you comments. 

Reviewer 2 Report

Comments and Suggestions for Authors

This is a very good systematic review. The following are my kind and subtle suggestions for reference.

1. In line 97, the selection criteria had a “NOT (laboratory).” Was it a typo? If not, it conflicts with Table 1 about the study design for “Experimental.”

2. In lines 337-338, a reference seemed missed. “This trend was observed in studies conducted in low middle income countries [ref ].”

3. For 3.3. Preventive pathway characteristics, including 3.3.1-3.3.4 sections, and 3.4. section, it would be more readable to compare the 14 studies with a new Table. “The Appendix A” seemed too complicated. Another new and brief table would make the manuscript cited many more times in the future.

4. After the discussion section, could it be possible for the authors to try to offer a more concrete preventive program based on your findings in any way, table, figure, or the like? It will be easier to make a similar one for future practitioners.

Author Response

(The authors gave the same response as above.)

Round 2

Reviewer 1 Report

Comments and Suggestions for Authors

Compliments to the Authors for the revised paper. Please, see the attachment for minor final adjustments.

Comments on the Quality of English Language

English language is good. Some little parts could be however improved

Author Response

Review Report Manuscript ID: geriatrics-3394831-v2

The Authors did a great work for addressing the suggested revisions. The paper is  greatly improved, clarified and integrated with useful and important infos. Only  some little/minor issues/inaccuracies are still present and easy to manage.

Authors

We would like to thank the reviewer for appreciating our work. We revised the text to address the minor issues.

Materials and Methods.

  • Par. 2.3. Data extraction:

o Line 116: “…. titles and abstract screening was performed with Microsoft Excel…”. The red part seems missing in the sentence of the Authors.

Authors

We revised it.

Par. 2.4. Risk of bias assessment:

  • Line 129: “…. applied the Joanna Briggs Institute JBI…”. I would suggest to put in brackets (JBI).

Authors

We revised it.

Results.

  • Par. 3.2. Risk of bias results.

o Line 180: “Both the qualitative studies [23,26] obtained a medium quality rating”. In table 3, item n. 26 has a high quality (not medium).

Authors

We revised the text.

“The qualitative study performed by Green et al.[23] obtained medium quality quality rating due to the lack of clarity regarding the researcher’s cultural or theoretical positioning while the other qualitative study [26] was scored as high”

  • Par. 3.3. Preventive pathway characteristics.

o Line 216: “All the 12 studies [18–22,24,25,27–30] …”. Records in brackets are 11 and not 12, as stated.

Authors

We apologize for the mistake, we updated the reference.

 [18-25, 27-30]

Supplementary Materials: line 420, regarding PRISMA checklist, I see only a table in the Supplementary pdf (1.5, i.e., S5). Maybe the table at p. 4 is S6.

Authors

We apologize for the mistakes, we revised the number.

Minor issues. For embedded citations in the text with pagination, please, add the page numbers, e.g.; line 37:

[3] (p......); line 45: [4] (p……).

Authors

We understand your concern however we follow the guideline of the journal to manage citation within the text. If some changes are required the editorial board will tell us to modify the reference before the visual version of the work.